Artemisinin mimics calorie restriction to trigger mitochondrial biogenesis and compromise telomere shortening in mice

Wang Da-Ting 1
He Jiang 1
Wu Ming 2
Li Si-Ming 3
Gao Qian 1
Zeng Qing-Ping 1 qpzeng@163.com
1 Tropical Medicine Institute, Guangzhou University of Chinese Medicine , Guangzhou , China
2 School of Life Science, Sun Yat-sen University , Guangzhou , China
3 The Second Affiliated Hospital, Guangzhou University of Chinese Medicine , Guangzhou , China
Rocha Joao
Electronic publication date: 2015 Mar 5
Publication date: 2015
Volume: 3
Electronic Location ID: e822
Received 2014 Oct 30; Accepted 2015 Feb 16
Copyright: © 2015 Wang et al.
Copyright year: 2015
Copyright holder: Wang et al.
License: This is an open access article distributed under the terms of the Creative Commons Attribution License, which permits unrestricted use, distribution, reproduction and adaptation in any medium and for any purpose provided that it is properly attributed. For attribution, the original author(s), title, publication source (PeerJ) and either DOI or URL of the article must be cited.
License URL: https://creativecommons.org/licenses/by/4.0/

Keywords: Artemisinin, Calorie restriction, Nitric oxide, Hydrogen peroxide, BRCA1, Telomere

Funding: National Science Foundation of China (NSFC) Financial support was provided by the National Science Foundation of China (NSFC). The funders had no role in study design, data collection and analysis, decision to publish, or preparation of the manuscript.

==============================
Calorie restriction is known to extend lifespan among organisms by a debating mechanism underlying nitric oxide-driven mitochondrial biogenesis. We report here that nitric oxide generators including artemisinin, sodium nitroprusside, and L-arginine mimics calorie restriction and resembles hydrogen peroxide to initiate the nitric oxide signaling cascades and elicit the global antioxidative responses in mice. The large quantities of antioxidant enzymes are correlated with the low levels of reactive oxygen species, which allow the down-regulation of tumor suppressors and accessory DNA repair partners, eventually leading to the compromise of telomere shortening. Accompanying with the up-regulation of signal transducers and respiratory chain signatures, mitochondrial biogenesis occurs with the elevation of adenosine triphosphate levels upon exposure of mouse skeletal muscles to the mimetics of calorie restriction. In conclusion, calorie restriction-triggered nitric oxide provides antioxidative protection and alleviates telomere attrition via mitochondrial biogenesis, thereby maintaining chromosomal stability and integrity, which are the hallmarks of longevity.

Introduction

Calorie restriction (CR) is a robust and extensively reproducible intervention of lifespan extension among organisms ranging from yeast to mammals (Koubova & Guarente, 2005; Spindler, 2010). CR is supposed to exert a longevity-promoting effect through enhanced mitochondrial biogenesis, which is initiated by nitric oxide (NO) derived from endothelial nitric oxide synthase (eNOS) (Nisoli et al., 2003; Nisoli et al., 2005; López-Lluch et al., 2006). It is also noted that the increment of respiratory activity increases cell and animal longevity (Lanza & Nair, 2010). An ‘uncoupling to survival’ hypothesis suggests that CR may increase respiratory activity and extend life expectancy by uncoupling oxidation from phosphorylation (Brand, 2000). Indeed, several mitochondrial uncoupling strategies allow lifespan extension in yeast (Barros et al., 2004), nematodes (Lemere et al., 2009), and fruit flies (Humpherey et al., 2009). A low dose of the mitochondrial uncoupler 2,4-dinitrophenol (DNP) remarkably extends mouse lifespan (Cerqueira, Laurindo & Kowaltowski, 2011).

It is clear that DNP carries protons to leak across the inner mitochondrial membrane, leading to the disconnection of both adenosine triphosphate (ATP) regeneration from adenosine monophosphate (AMP) and oxidized nicotinamide adenine dinucleotide (NAD+) conversion to reduced nicotinamide adenine dinucleotide (NADH + H+) (Korde et al., 2005). The increases of AMP and NAD+ can separately activate AMP-activated kinase (AMPK) and NAD+-dependent deacetylase Sirtuin 1 (SIRT1), which can coordinately activate peroxisome proliferator-activated receptor-γ co-activator 1α (PGC-1α) essential for mitochondrial biogenesis (Rodgers et al., 2005; Lee et al., 2006). It has been recently demonstrated that the AMPK activator metformin mimics CR to improve healthspan and extend lifespan in mice (Martin-Montalvo et al., 2013). The SIRT1 activator resveratrol has been also known to exert CR-like beneficial effects on obese humans’ life quality (Blagosklonny, 2010).

Since metabolic suppression was suggested to mitigate DNA damage (Koubova & Guarente, 2005), a novel model deciphering CR-conferred DNA protection has been established, in which CR-mediated metabolic/hormonal adaptations result in cellular adaptations including reduced cell proliferation, increased autophagy or apoptosis, up-regulated DNA repair systems, and enhanced genomic stability (Longo & Fontana, 2010). Most recently, CR has been shown to synergize with telomerase for promoting mouse longevity, suggesting a role of shortened telomeres in aging (Vera et al., 2013). Nevertheless, it has not yet been identified the mechanism by which CR protects DNA and telomeres.

According to the findings that CR induces NO (Nisoli et al., 2003; Nisoli et al., 2005) and NO competitively binds to cytochrome c oxidase (COX) (Mason et al., 2006), we propose here that CR-triggered NO might interact with COX to initiate mitochondrial uncoupling, which would provoke oxidative burst, activate antioxidative responses, mitigate DNA damage, and thereby compromise telomere shortening. To provide evidence supporting our proposition, we choose three different types of in vivo NO generators to replicate the effect of CR-triggered NO on the integrity of telomeres in mice. Artesunate (ART) is a semi-synthetic soluble derivative of artemisinin, a sesquiterpene endoperoxide that has been clinically used for antimalaria, and has been identified as an inhibitor of nitric oxide synthase (NOS) and an inducer of NO (Zeng & Zhang, 2011; Zeng et al., 2011). Sodium nitroprusside (SNP) as an NO donor and L-arginine (ARG) as an NO precursor have been widely used in modern medicine. Additionally, hydrogen peroxide (H2O2) was also included to simulate NO-posed oxidative stress that elicits antioxidative responses.

From the present study, we have disclosed the implication of NO signaling in telomere maintenance, and replayed the molecular episode of NO-mediated telomere protection. Also, we have rehearsed H2O2-compromised telomere shortening. In such context, we can explain why CR extends lifespan by annotating that CR triggers NO burst, initiates mitochondrial biogenesis, scavenges reactive oxygen species, attenuates DNA damage, and alleviates telomere attrition, thereby eventually leading to lifespan extension. We expect that the elucidation of de novo mechanisms underlying anti-aging/longevity in mammals should beneficial to a better solution to more and more severe human health issues.

Materials and methods

Animals and treatment procedures

Kunming (KM) mice, belonging to an outbred population originated from SWISS mice, were used in the present study. All mice were housed on a 12-h light: 12-h dark cycle at 25 °C, and fed with either ad libitum (AL) or 60% AL (CR). For treatment, AL mice were injected by 260 µM ART, 67 µM SNP, 5.7 mM ARG, or 200 µM H2O2 in 50 µl injection volume/20 g body weight. Each drug was injected into an identical loci of the skeletal muscles on one hind-leg, and samples were collected from skeletal muscle tissues around the injected sites. Animal procedures were in accordance with the animal care committee at the Guangzhou University of Chinese Medicine, Guangzhou, China. The protocol was approved by the Animal Care Welfare Committee of Guangzhou University of Chinese Medicine (Permit Number: SPF-2011007).

Enzyme-linked immunosorbent assay (ELISA)

All target proteins/peptides, including CAT, COX4, eNOS, GSH, SIRT3, SOD1, and SOD2, as well as the reference protein, glyceraldehyde-3-phosphate dehydrogenase (GAPDH), were immunoquantified according to antibody manufacture’s manuals. The antibody against eNOS was purchased from Assay Biotechnology Co. Ltd. (Sunnyvale, CA, USA). The antibody against COX4 was purchased from Beijing Biosynthesis Biotechnology Co. Ltd. (Beijing, China). Other first antibodies were purchased from R & D Systems, Inc. ROS levels were measured with a Mouse ROS ELISA Kit (EIAab Science Co. Ltd., Wuhan, China) following the manufacturer’s instructions.

Western blotting

The antibodies against AMPKα1/2 (H-300), AMPKα1/2 (Thr172), Akt1 (B-1), CYT C (H-104), PGC-1 (H-300), p-Akt1/2/3 (Ser473), and p-MFN2 (H-68) were purchased from Santa Cruz Biotechnology, Inc. (Dallas, Texas, USA). The SIRT1 antibody was purchased from Milipore (Temecula, California, USA). The eNOS antibody was purchased from Assay Biotechnology Co. Ltd. (Sunnyvale, CA, USA). The p-eNOS (Ser1177) antibody was purchased from Cell Signaling Technology, Inc. (Danvers, Massachusetts, USA). The blotting experiments were performed obeying to the manufacturer’s instruction manuals. The gray scale values from blotted proteins were measured using a scanning instrument, and the raw data of gray scale values were normalized by a gray scale value of the reference protein GAPDH available from each group.

Southern blotting

Total DNA was extracted and digested by the restriction enzymes Hinf I and Rsa I. Southern blotting was performed by the hybridization of digested DNA with digoxin-labeled (TTAGGG)4 probes according to the manufacturer’s instructions. The lengths of telomere fragment bands on the gel were estimated by comparison with the base pairs of DNA standards.

Quantitative polymerase chain reaction (qPCR)

Total RNA was extracted from mouse skeletal muscle cells by a Trizol methods. The primers with the sequences listed in Table 1 were synthesized by Invitrogen (Carlsbad, California, USA). The copy numbers of amplified genes were estimated by 2−ΔΔCt, in which ΔΔCt = [target gene (treatment group)/target gene (control group)]/[house-keeping gene (treatment group)/house-keeping gene (control group)]. The raw qPCR data were normalized by the copy numbers of the reference gene GAPDH.

Table 1 The primer sequences and fragment lengths of amplified genes.

Genes	Primers	Fragment lengths (bp)	
GAPDH	F:5’GTTGTCTCCTGCGACTTCA3’
R:5’GCCCCTCCTGTTATTATGG3’	293	
BRCA2	F:5’AAGCCAAGCCACATAGCACAG3’
R:5’ACTCCAGCCGAACCTTCAAAT3’	164	
RB	F:5’AAATCATCGTCACTGCCTACAA3’
R:5’AGGAATCCGTAAGGGTGAACT3’	236	
MYC	F:5’GACTGTATGTGGAGCGGTTTCT3’
R:5’GCTGTCGTTGAGCGGGTAG3’	213	
RAD50	F:5’AGTTTACTCCCAGTTCATTACGC3’
R:5’CTCTATTGACACTCTGTAGTCGGTT3’	287	
RAD51	F:5’CTGCCCTTTACAGAACAGACTACTC3’
R:5’GGCTACTACCTGGTTGGTGATTAC3’	140	
TERT	F:5’GCCCAGACCTCAATTAAGACGA3’
R:5’CTTCAACCGCAAGACCGACA3’	96	

RT-PCR array

The Mouse Ubiquitylation Pathway RT2 Profiler™PCR Array was provided by SABioscience Qiagen, (Hilden, Germany). The experiments were performed by Kangchen Biotechnology Co., Ltd., Shanghai, China.

Laser confocal microscopy

The fresh samples of mouse skeletal muscles were fixed in paraformadehyde for 24 h. After repeatedly rinsed, the fixed tissues were dehydrated by gradient ethanol. For embedding and sectioning, the tissue slices were pasted on the slides and coated at 50 °C. After they were dewaxed by a transparent reagent and rinsed, the slides were incubated with antibodies and stained by DAPI. After drying, a fluorescence quencher was added and slides were sealed. The fluorescence-labeled second antibodies and the first antibodies against BRCA1 and TERT were purchased from Beijing Biosynthesis Biotechnology Co., Ltd. (Beijing, China). Immunoblotting was carried out based on the manufacturer’s instructions.

Electronic microscopy

After treatment, cells were harvested and fixed in 2.5% glutaraldehyde in 0.1 M phosphate buffer for three hours at 4 °C, followed by post-fixation in 1% osmium tetroxide for one hour. Samples were dehydrated in a graded series of ethanol baths, and infiltrated and embedded in Spurr’s low-viscosity medium. Ultra-thin sections of 60 nM were cut in a Leica microtome, double-stained with uranyl acetate and lead acetate, and examined in a Hitachi 7700 transmission electron microscope at an accelerating voltage of 60 kV.

Determination of NO and ATP levels

The NO and ATP levels were determined by the reagent kits manufactured by Jiancheng Biotechnology Institute, Nanjing, China. The NO levels (µM/g) = (ODtest − ODblank)/(ODstandard − ODblank) • standard nitrate concentration (20 µM)/sample protein concentration (µg/µl). The ATP levels (µM/g) = (ODtest − ODblank)/(ODstandard − ODblank) • standard ATP concentration (1000 µM) • sample dilution folds/sample protein concentration (µg/µl).

Statistical analysis

Statistical analyses were conducted by the one-way ANOVA method using SPSS version 17.0 for Windows. All data were represented as mean ± SEM unless otherwise stated. The XY graphs and column graphs were plotted and depicted using GraphPad Prism version 4.0.

Results

CR and mimetics up-regulate eNOS and COX4 coordinately in a time-dependent manner

Considering that the up-regulation of mitochondrial genes is essential for CR-elicited mitochondrial biogenesis via NO signaling, we tried to validate whether CR might up-regulate the mitochondrial biomarker COX4 via enhanced eNOS expression. For this purpose, we determined the quantities of eNOS and COX4 in the skeletal muscles of mice exposed to CR or injected by ART, SNP, or ARG. Because NO-driven mitochondrial biogenesis was assumed to accompany with oxidative burst, we also used H2O2 to mimic CR’s inducible effects on the expression of eNOS and COX4.

Like CR exposure for as long as three months, treatment of mice by ART, SNP, ARG, or H2O2 for one, three, six hours, or three days allows the gradual increases of both eNOS quantities (Fig. 1A) and COX4 quantities (Fig. 1B). To reach a level of significant difference from the control (AL mice), ARG needs only one hour, H2O2 needs three hours, SNP needs six hours, and ART needs even three days. Among groups, ARG treatment exhibits the largest quantities of eNOS and COX4, even larger than those upon exposure to CR. Interestingly, H2O2 treatment also induces larger quantities of COX4 and equal quantities of eNOS as compared with NO generators.

Figure 1 ART, SNP, ARG, or H2O2 mimics CR to increase the quantities of eNOS and COX4 in mouse skeletal muscle cells.

(a) ELISA for eNOS measurement after treatment for different durations. (b) ELISA for COX4 measurement after treatment for different durations. 0 h represents AL; 3 m indicates CR for three months; and 1 h, 3 h, or 6 h means treatment for one hour, three hours, or six hours. ART (260 µM), SNP (67 µM), ARG (5.7 mM), or H2O2(200 µM) was injected into the mouse skeletal muscle in the dose of 50 µl volume/20 g body weight. The 1 h, 3 h, or 6 h group had only one injection, and the 3d group had three daily injections. The ages of mice used are two-month-old except for CR mice, which are four-month-old with one-month AL and three-month CR treatment. The significance of statistical difference between a treatment sample and the AL sample was represented by * P < 0.05; ** P < 0.01; *** P < 0.001(n = 3).

These results indicate that ART, SNP, ARG, or H2O2 can mimic CR to induce the enhanced expression of eNOS and COX4, implying that CR may affect mitochondrial structure and enhance mitochondrial function through the involvement of NO and H2O2. Because accompanying with NO-posed oxidative stress, CR and mimetics are anticipated to elicit antioxidative responses at least in mitochondria or even throughout whole cells.

CR and mimetics attenuate oxidative stress upon eliciting antioxidative responses

To reveal the effects of CR-triggered NO and H2O2 on the oxidative and antioxidative homeostasis, we monitored the dynamic changes of mitochondrial manganese superoxide dismutase (Mn-SOD, SOD2) and its activator SIRT3 in the skeletal muscle cells of mice treated by CR and mimetics. Consequently, both Mn-SOD (Fig. 2A) and SIRT3 (Fig. 2B) are synchronously up-regulated in a time-dependent manner after exposure to CR or injection by ART, SNP, ARG, or H2O2. While ARG induces the highest Mn-SOD quantity, SNP induces the highest SIRT3 quantity. These results demonstrate that antioxidation against oxidation is initiated from the activation of SIRT3-SOD2 in mitochondria of skeletal muscle cells after treatment of mice by CR and mimetics.

Figure 2 ART, SNP, ARG, H2O2, or CR activates antioxidant networks for ROS scavenging.

(A) and (B) ELISA measurement of time-dependently induced mitochondria-localized Mn-SOD and SIRT3 by CR and mimetics. (C), (D), and (E) ELISA measurement of time-dependently induced SOD, CAT, and GSH by CR and mimetics. (F) ELISA measurement of the total ROS level in mice treated by CR and mimetics. 0 h represents AL; 3 m indicates CR for three months; and 1 h, 3 h, or 6 h means treatment for one hour, three hours, or six hours. ART (260 µM), SNP (67 µM), ARG (5.7 mM), or H2O2(200 µM) was injected into the mouse skeletal muscle in the dose of 50 µl volume/20 g body weight. The 1 h, 3 h, or 6 h group had only one injection, and the 3d group had three daily injections. The ages of mice used are two-month-old except for CR mice, which are four-month-old with one-month AL and three-month CR treatment. The significance of statistical difference between a treatment sample and the AL sample was represented by * P < 0.05; ** P < 0.01; *** P < 0.001 (n = 3).

Except for specifically up-regulating Mn-SOD, CR and mimetics were also found to increase the quantities of total SOD enzymes that include cytosolic copper/zinc SOD (Cu/Zn-SOD, SOD1) (Fig. 2C). Additionally, CR and mimetics-treated mice exhibit the increases of catalase (CAT) and glutathione (GSH) (Figs. 2D and 2E), among which ARG induces the highest levels of SOD and GSH, while H2O2 induces the highest level of CAT. These results indicate that CR and mimetics can coordinately activate the antioxidative network in skeletal muscle cells, at least including SOD, CAT, and GSH.

Accordingly, a significant decline of the total levels of reactive oxygen species (ROS) in mouse skeletal muscle cells was observed after treatment of mice by ART, SNP, ARG, H2O2, or CR (Fig. 2F), suggesting an essential consequence of ROS scavenging by activated antioxidant enzymes. After treatment of mice by CR mimetics for one hour, SNP renders the lowest ROS level, H2O2 confers the second lower ROS level, and ARG and ART keeps a relatively lower ROS level than AL.

These results demonstrate that CR and mimetics can effectively stimulate the antioxidative responses in mouse skeletal muscle cells to quench ROS and create a less oxidative stress milieu.

CR and mimetics mostly down-regulate ubiquitylation pathway genes including tumor suppressors responsible for DNA repair

To make sure the relevance of CR and mimetics to the ubiquitin-mediated proteolysis pathway (UMPP) that is involved in the auto-regulated degradation of proteins, we set out to investigate whether CR and mimetics would affect the expression of ubiquitylation pathway genes. From the transcript profiling of ubiquitylation genes among ART, SNP, ARG, H2O2, and CR groups, it was noted that all 84 ubiquitylation genes examined are mostly down-regulated (Fig. 3, and see also Tables S1–S5 for details).

Figure 3 A hierarchical clustering illustration for the up/down-regulation of 84 ubiquitylation genes from RT-PCR array data.

The red color represents up-regulation as compared with AL; and the green color represents down-regulation as compared with AL. The RT-PCR array was performed after daily injection for three days into mouse skeletal muscle by 260 µM ART, 67 µM SNP, 5.7 mM ARG in the dose of 50 µl volume/20 g body weight, or one injection by 200 µM H2O2(50 µl/20 g), and sampling after last injection for six hours. The ages of all mice used are four-month-old, among which CR mice have one-month AL and three-month CR treatment.

Among examined genes, 11 genes encoding ubiquitin-activating enzyme (E1) are unchanged or down-regulated in different groups, whereas 73 genes encoding ubiquitin-conjugating enzyme (E2) and ubiquitin-protein ligase (E3) genes are mostly down-regulated at a different extent. For example, one of the autophagy genes, ATG7 (E1), is down-regulated for 35 folds in CR and for 2–5 folds in other treatment groups, and UBE2C (E2/E3) is down-regulated for 112 folds in CR and for 16–45 folds in other treatment groups. These results suggest that ubiquitylation-tuned protein degradation has been compromised upon exposure to CR for three months or after treatment by CR mimetics for three days.

The analysis of RT-PCR array data also indicates that a few of tumor suppressors and some DNA repair proteins are down-regulated, in which BRCA1 is down-regulated for approximately 25 folds among all treatment groups, while BARD1 is down-regulated for 30–60 folds by ART, SNP, or ARG, and five folds by CR. TRP53 is slightly down-regulated in all treatment groups. Furthermore, we also detected the down-regulation of some BRCA1 partner-encoding genes including BRCA2, MYC, RAD50, and RAD51 in CR mice. While RB is unchanged in CR mice, MYC and RB are mildly up-regulated by ART and H2O2. Besides, we observed that telomerase reverse transcriptase gene (TERT) is down-regulated by ARG, CR, and SNP, and unchanged after treatment by ART and H2O2 (Table 2).

Table 2 Quantification of amplified transcripts from DNA repair genes and TERT in skeletal muscle cells of mice treated by CR, H2O2, ART, SNP, ARG, or AL.

(A) The normalization of quantified transcripts of DNA repair genes and TERT by a specific target gene vs the reference gene GAPDH. (B) The fold changes of quantified transcripts of DNA repair genes and TERT in treated mice vs AL mice.

Sample	BRCA2
/GAPDH	RB
/GAPDH	MYC
/GAPDH	RAD50
/GAPDH	RAD51
/GAPDH	TERT
/GAPDH	
CR	6.48E−05	3.21E−04	6.40E−05	3.33E−05	9.68E−05	1.64E−05	
H2O2	9.87E−05	1.82E−03	4.62E−03	2.95E−04	1.39E−04	3.07E−05	
ART	9.29E−05	1.48E−03	1.56E−03	5.79E−04	1.87E−04	4.14E−05	
SNP	1.36E−04	1.64E−03	3.90E−04	3.31E−04	1.83E−04	1.94E−05	
ARG	6.51E−05	1.17E−03	2.71E−04	3.02E−04	1.50E−04	7.36E−06	
AL	1.13E−04	3.00E−04	2.83E−04	2.33E−04	1.29E−04	2.87E−05	
Comparison	BRCA2
/GAPDH	RB
/GAPDH	MYC
/GAPDH	RAD50
/GAPDH	RAD51
/GAPDH	TERT
/GAPDH	
CR/AL	0.57	1.07	0.23	0.14	0.75	0.57	
H2O2/AL	0.87	6.07	16.33	1.27	1.08	1.07	
ART/AL	0.82	4.93	5.51	2.48	1.45	1.44	
SNP/AL	1.20	5.47	1.38	1.42	1.42	0.68	
ARG/AL	0.58	3.90	0.96	1.30	1.16	0.26	
Notes.

The RT-PCR was performed after daily injection for three days into mouse skeletal muscle by 260 µM ART, 67 µM SNP, 5.7 mM ARG in the dose of 50 µl volume/20 g body weight, or one injection by 200 µM H2O2 (50 µl/20 g), and sampling after last injection for six hours. The ages of all mice used are four-month-old, among which CR mice have one-month AL and three-month CR treatment.

From the fact that DNA repair genes are mostly down-regulated by CR and mimetics, we conclude that DNA damage in mice treated by CR and mimetics should be attenuated in a low oxidative milieu that is resulted from antioxidative activation and ROS scavenging. It is also conclusive that because oxidative DNA lesions are mitigated, TERT is of course accordingly down-regulated or unchanged in all treatment groups.

CR and mimetics-maintained longer telomeres are correlated with suppressed oxidative circumstance

From the results regarding the global down-regulation of tumor suppressor genes and accessory DNA repair genes by CR and mimetics, it can be expected that mice treated by CR and mimetics should have longer telomeres. To confirm this deduction, we compared the lengths of telomere restriction fragments (TRFs) from the skeletal muscle cells of CR and mimetics-treated mice to those of an AL mouse. Consequently, TRFs of an AL sample were found to shift faster than those of ART, SNP, ARG, H2O2, and CR samples on the gel, suggesting AL TRFs being shorter than ART, SNP, ARG, H2O2, and CR TRFs (Fig. 4).

Figure 4 Hybridization detection of TRFs in mouse skeletal muscle cells of AL, CR and mimetics-treated mice.

For Southern blotting, samples were collected from the skeletal muscle of AL and CR mice, or from CR mimetics mice injected by 260 µM ART, 67 µM SNP, 5.7 mM ARG, or 200 µM H2O2 in the dose of 50 µl volume/20 g body weight for three times, in which the 1st, 2nd, and 3rd injections are on the 1st, 3rd, and 5th day, respectively. The ages of AL and CR mimetics-treated mice are two-month-old, and the ages of CR mice are five-month-old, including one-month AL and four-month CR treatment.

For more accurate comparison of TRFs among groups, we further measured the main band lengths, the longest band lengths, and the shortest band lengths of TRFs, and accounted for their average band lengths, as listed in Table 3. It is clear that SNP and H2O2 render longer average band lengths, whereas AL and CR confer shorter average band lengths. The main band lengths can be sorted as SNP > H2O2 > ARG > ART > CR > AL, and the longest band lengths are in the order of SNP > CR/H2O2 > ARG > AL > ART.

Table 3 Measurement of TRF lengths in mouse skeletal muscle cells among AL, CR, and CR mimetics groups.

Group	The main band length (bp)	The longest band length (bp)	The shortest band length (bp)	The average band length (bp) x¯±s	
AL	2353	3077	1592	2341 ± 743	
ART	2450	2956	2058	2488 ± 450	
SNP	2917	3917	1717	2850 ± 1102	
ARG	2551	3566	1476	2531 ± 1045	
H2O2	2691	3814	1717	2741 ± 1049	
CR	2450	3814	1142	2469 ± 1336	
Notes.

For TRF measurement, samples were collected from the skeletal muscles of AL and CR mice, or from CR mimetics mice injected by 260 µM ART, 67 µM SNP, 5.7 mM ARG, or 200 µM H2O2 in the dose of 50 µl volume/20 g body weight for three times every other day. The ages of AL and CR mimetics mice are two-month-old, but the ages of CR mice are five-month-old, including one-month AL treatment and four-month CR treatment.

Interestingly, SNP and H2O2 lead to the lowest ROS levels (see Fig. 2F), which may partly decipher why SNP and H2O2 allow longer telomeres because longer telomeres are correlated with few ROS and less DNA damage. In contrast, AL and CR show the higher levels of ROS (see also Fig. 2F), thus providing an explanation on the relevance of more ROS to shorter telomeres. Why does CR give rise to the shortest TRFs than CR mimetics and even AL? This is likely due to the older ages of CR mice because they were older than other mice by three months when the telomere lengths were measured. As to the reason why younger mice were chosen for treatment by CR mimetics, we consider that the shortened telomeres in older mice should not be extended by CR mimetics.

Co-existence of BRCA1 and TERT in similar abundance in nuclei implies an interaction of BRCA1 with TERT

As described above, BRCA1 and TERT are down-regulated at the level of transcription (the mRNA level) (see Fig. 3 and Table 2). To ensure if BRCA1 and TERT are also down-regulated at the level of translation (the protein level), we tried to phenotyping the localization of BRCA1 and TERT in the skeletal muscle cells of AL, CR, and CR mimetics-treated mice. As a consequence, TERT was shown to co-exist with BRCA1 in overlapped nuclear locations, which can be clearly observed from the AL sample (Fig. 5), suggesting that TERT and BRCA1 may be interactive and cooperative. As to the dimmed BRCA1-TERT signals in ART, SNP, and ARG samples, they might represent the dual down-regulation of BRCA1 and TERT.

Figure 5 Laser confocal microscopic phenotyping of coordinated down-regulation of BRCA1 with TERT in mouse skeletal muscles treated by ART, SNP, ARG, H2O2, or CR.

Green fluorescence indicates BRCA1, red fluorescence indicates TERT, and blue fluorescence represents 4’,6-diamidino-2-phenylindole (DAPI)-staining nuclear DNA. For observation by a laser confocal microscope, samples were collected from the skeletal muscles of mice injected by 260 µM ART, 67 µM SNP, 5.7 mM ARG or 200 µM H2O2 (50 µl/20 g) for three times, in which the 2nd and the 3rd injections are on the 3rd and 5th day, respectively. The ages of AL and CR mimetic mice are two-month-old, but the ages of CR mice are five-month-old, including one-month AL treatment and four-month CR treatment.

Because the fluorescence strengths of TERT and BRCA1 are almost identical in each group albeit with lighter or darker fluorescence due to up- or down-regulation, we assume that both DNA-maintaining proteins perhaps accumulate with similar abundance, which is likely tuned by oxidative-antioxidative homeostasis.

ART, SNP, or ARG up-regulates eNOS, upstream protein kinases, and downstream respiratory biomarkers

To ascertain the possibility of ART, SNP, or ARG mediating NO signaling, we evaluated the expression and phosphorylation of eNOS and its upstream protein kinases, including Akt and AMPK. As results, their non-phosphorylated/phosphorylated forms, AMPK and p-AMPKThr172, Akt and p-AktSer473, and eNOS and p-eNOSSer1177, are simultaneously induced in the skeletal muscle cells of mice injected by ART, SNP, or ARG (Fig. 6A and Table 4). AMPK and p-AMPKThr172 exhibit almost identical expression levels, suggesting a synchronous mode of AMPK expression and phosphorylation. Akt and eNOS show higher levels than p-AktSer473 and p-eNOSSer1177, implying only a minor of Akt and eNOS being phosphorylated. These results indicate that ART, SNP, or ARG can synchronously induce eNOS, Akt, and AMPK, and partially activate them into p-eNOSSer1177, p-AktSer473, and p-AMPKThr172.

Figure 6 Western blotting of target proteins in mouse skeletal muscles injected by ART, SNP, or ARG.

(A) Up-regulation and phosphorylation of eNOS and upstream protein kinases. (B) Up-regulation of mitochondrial biomarkers and relevant signal transducers. (C) The time-course mode of up-regulation of signal transducers and mitochondrial biomarkers. Western blotting was performed after three days of daily injection by 260 µM ART, 67 µM SNP, or 5.7 mM ARG (50 µl volume/20 g body weight). For each group of blots, only one stripe of gel with GAPDH bands was shown as reference, but blotting of each target protein was parallelly performed with GAPDH for comparison.

Table 4 The time-course monitoring of expression levels of eNOS and upstream/downstream target proteins in the skeletal muscle cells of mice treated by ART, SNP, or ARG.

(A) The gray scale values for target proteins to the reference protein GAPDH. (B) The fold changes of gray scale values for target proteins in treated mice to AL mice.

Target/reference protein	ART	SNP	ARG	AL	
AMPK/GAPDH	1.01 ± 0.01*	0.97 ± 0.23**	0.85 ± 0.08*	0.61 ± 0.09	
p-AMPK/GAPDH	1.00 ± 0.02**	0.93 ± 0.04*	0.93 ± 0.01*	0.83 ± 0.01	
Akt/GAPDH	0.97 ± 0.01**	0.95 ± 0.02**	0.62 ± 0.08*	0.31 ± 0.06	
p-Akt/GAPDH	0.56 ± 0.40	0.21 ± 0.11	0.13 ± 0.11	0.09 ± 0.08	
eNOS/GAPDH	0.93 ± 0.10***	0.94 ± 0.01***	0.97 ± 0.02***	0.15 ± 0.01	
p-eNOS/GAPDH	0.86 ± 0.11*	0.53 ± 0.37	0.21 ± 0.06	0.19 ± 0.16	
SIRT1/GAPDH	1.01 ± 0.05**	0.90 ± 0.03**	0.80 ± 0.05*	0.62 ± 0.04	
PGC-1α/GAPDH	1.04 ± 0.63**	1.00 ± 0.04**	0.99 ± 0.05**	0.26 ± 0.16	
MFN2/GAPDH	0.93 ± 0.02	0.58 ± 0.38	0.68 ± 0.03	0.32 ± 0.24	
CYT C/GAPDH	0.94 ± 0.01**	0.67 ± 0.18*	0.50 ± 0.41*	0.11 ± 0.04	
3 h AMPK/GAPDH	1.27 ± 0.05	1.24 ± 0.05	1.09 ± 0.05	0.77 ± 0.27	
6 h AMPK/GAPDH	1.05 ± 0.22	0.95 ± 0.27	0.93 ± 0.31	0.68 ± 0.14	
24 h AMPK/GAPDH	1.05 ± 0.27	1.06 ± 0.12	0.85 ± 0.04	0.69 ± 0.12	
3 h PGC-1α/GAPDH	1.07 ± 0.03**	0.94 ± 0.06*	0.90 ± 0.02*	0.75 ± 0.03	
6 h PGC-1α/GAPDH	0.79 ± 0.14*	0.75 ± 0.01	0.58 ± 0.01	0.34 ± 0.15	
24 h PGC-1α/GAPDH	1.30 ± 0.07**	1.16 ± 0.03**	1.06 ± 0.07	0.58 ± 0.11	
3 h CYT C/GAPDH	1.13 ± 0.02*	0.95 ± 0.02	0.93 ± 0.17	0.76 ± 0.02	
6 h CYT C/GAPDH	1.29 ± 0.04**	1.01 ± 0.07*	0.97 ± 0.17*	0.64 ± 0.03	
24 h CYT C/GAPDH	1.40 ± 0.03**	1.25 ± 0.05**	0.71 ± 0.22	0.53 ± 0.06	
Treated mice/AL mice	ART/AL	SNP/AL	ARG/AL	
AMPK/GAPDH	1.66	1.59	1.39	
p-AMPK/GAPDH	1.20	1.12	1.12	
Akt/GAPDH	3.13	3.06	2.00	
p-Akt/GAPDH	6.22	2.33	1.44	
eNOS/GAPDH	6.20	6.27	6.47	
p-eNOS/GAPDH	4.53	2.79	1.11	
SIRT1/GAPDH	1.63	1.45	1.29	
PGC-1α/GAPDH	4.00	3.85	3.81	
MFN2/GAPDH	2.91	1.81	2.13	
CYT C/GAPDH	8.55	6.09	4.55	
3 h AMPK/GAPDH	1.65	1.61	1.42	
6 h AMPK/GAPDH	1.54	1.40	1.37	
24 h AMPK/GAPDH	1.52	1.54	1.23	
3 h PGC-1α/GAPDH	1.43	1.25	1.20	
6 h PGC-1α/GAPDH	2.32	2.21	1.71	
24 h PGC-1α/GAPDH	2.24	2.00	1.83	
3 h CYT C/GAPDH	1.49	1.25	1.22	
6 h CYT C/GAPDH	2.02	1.58	1.52	
24 h CYT C/GAPDH	2.64	2.36	1.34	
Notes.

Western blotting was performed after daily injection for three days by 260 µM ART, 67 µM SNP, or 5.7 mM ARG (50 µl volume/20 g body weight). GAPDH: glyceraldehyde-3-phosphate dehydrogenase. For each group of blots, only one stripe of gel with GAPDH bands was shown as reference, but blotting of each target protein was parallelly performed with GAPDH for comparison. The fold changes of gray scale values were calculated by comparing each target protein with corresponding GAPDH.

* The significance of statistical difference between a treatment sample and AL is represented by P < 0.05.

** The significance of statistical difference between a treatment sample and AL is represented by P < 0.01.

*** The significance of statistical difference between a treatment sample and AL is represented by P < 0.001 (n = 3).

To reinforce the relevance of NO-induced gene expression to mitochondrial biogenesis, we quantified some related signal transducers and mitochondria-targeted proteins in mouse skeletal muscles injected by ART, SNP, or ARG. Consequently, it was found that ART, SNP, or ARG leads to the significant up-regulation of the signaling components, SIRT1 and PGC-1α, and mitochondrial biomarkers, MFN2 and CYT C. As noted, PGC-1α shows the highest expression level, and SIRT1 also exhibits a mildly induced level (Fig. 6B and Table 4). These results indicate that NO can up-regulate the mitochondria-localized MFN2 and CYT C through inducing the mitochondrial biogenesis-necessitated SIRT1 and PGC-1α.

To understand the sequential events occurring in NO signaling and mitochondrial biogenesis, we followed up the time-course changes of selective signal transducers and mitochondrial biomarkers. In monitoring the expression of AMPK, PGC-1α, and CYT C in mouse skeletal muscles injected by ART, SNP, or ARG for three, six, and 24 h, we observed that AMPK reaches its maximal level within three hours and subsequently maintains a stable-steady level. PGC-1α and CYT C also exhibit the time-dependent expression manners, namely the three hour-treatment allows only the lower levels, the six hour-treatment leads to the elevated levels, and the 24 h-treatment gives rise to the highest levels. Importantly, the 24 h-expression level of PGC-1α is higher than that of CYT C (Fig. 6C and Table 4). These results demonstrate that the induction of understudied genes occurs in the sequelae from AMPK to PGC-1α and CYT C rather than vice versa.

CR mimetics-derived high-level NO predisposes mitochondrial biogenesis in mouse skeletal muscle cells

The ELISA and Western blotting data have revealed the induced up-regulation of eNOS by CR mimetics, but direct evidence confirming the elevation of NO levels is still lacking. We monitored the NO levels in skeletal muscles of mice injected by ART, SNP, ARG, or H2O2. A NO burst was seen after treatment for six hours although a decline trend was observed after treatment for three days (Fig. 7A), addressing that all kinds of CR mimetics used in this study play their roles upon NO signaling. Furthermore, we also measured the ATP levels in the skeletal muscles of mice injected by ART, SNP, ARG, or H2O2. The results as depicted in Fig. 7B indicate that ATP is increased after treatment for six hours, but maintains a steady-state higher level after treatment for three days. These results provide support to the assumption of CR mimetics-enhanced mitochondrial functionality.

Figure 7 Determination of NO and ATP levels and electronic microscopic phenotyping of mitochondria in mouse skeletal muscles injected by ART, ARG, SNP, or H2O2.

(A) The elevation of NO levels upon treatment by CR mimetics. (B) The elevation of ATP levels upon treatment by CR mimetics. (C)–(G) Mitochondrial density and structure in ART, ARG, SNP, H2O2, and AL, respectively. Samples were collected from mouse skeletal muscles after 6 h by one injection or by daily injection for three days by 260 µM ART, 67 µM SNP, 5.7 mM ARG, or 200 µM H2O2 (50 µl injection volume/20 g body weight).

At last, we scrutinized whether the density of mitochondria would be changed in mouse skeletal muscle cells exposed to ART, SNP, ARG, or H2O2. As compared with one-layer and linear-arrayed mitochondria in AL-exposed cells (Fig. 7C), SNP-treated cells (Fig. 7D) or H2O2-treated cells (Fig. 7E) show remarkable mitochondrial proliferation with multi-layer mitochondria, and ART-treated cells (Fig. 7F) or ARG-treated cells (Fig. 7G) also possess more mitochondrial layers than AL-exposed cells after treatment for six hours.

These results unambiguously indicate that CR and mimetics can produce NO, drive mitochondrial biogenesis, and recover energy supply in mice during a short period, for example, within six hours as examined in the present study.

Discussion

The mechanisms underlying CR-mediated lifespan extension have been eagerly and extensively investigated in recent years. A current research work in nematodes has well deciphered the reason why CR decreases ATP by reporting that increase of the citrate cycle intermediate α-ketoglutarate during CR exposure targets the subunit β of ATP synthase (complex V) and inhibits its activity, addressing an important role of mitochondrial uncoupling in prolonging lifespan (Chin et al., 2014). Although whether mitochondrial uncoupling correlates with redox homeostasis remains largely unknown, evidence is emerging to support the concept of mitochondrial hormesis (mitohormesis), which suggests that potent ROS burst from mitochondria evokes antioxidative responses and promotes life expectancy (Ristow & Kim, 2010; Ristow, 2014).

Indeed, mitochondrial superoxide production was found to increase the longevity of nematodes by triggering ROS-scavenging responses (Yang & Hekimi, 2010). It has been recently indicated that aspirin promotes mitochondrial biogenesis through H2O2 production and SIRT1/PGC-1α induction in cultured mouse liver cells (Kamble et al., 2013). A most new report has also demonstrated that H2O2 enables the up-regulation of mitochondria-specific SIRT3 and Mn-SOD in mice (Qiu et al., 2010). Following the finding of H2O2-mediated extension of yeast chronological lifespan through inducing antioxidative responses (Mesquita et al., 2010), we have also confirmed the mitohormetic effects of H2O2 on yeast chronological lifespan (Wang & Zeng, 2014). Most recently, metformin has been proven to promote lifespan in nematodes via the peroxiredoxin PRD-2-involved mitohormesis (De Haes et al., 2014).

In the present study, we found that ART, SNP, ARG, or H2O2 can mimic CR to induce SOD, CAT, and GSH, which lead to the alleviation of ROS-engaged stress. In particular, we observed the synchronous induction of mitochondrial SOD2 and its activator SIRT3 by CR and mimetics, which is in consistence with the known fact that CR dramatically reduces oxidative stress by inducing SIRT3-activated SOD2 (Qiu et al., 2010). Although it is understandable that H2O2 as an oxidant enables the induction of antioxidant enzymes, why ART, SNP, and ARG also induce antioxidant enzymes seems puzzling. In our opinion, there may be two possibilities: one is the direct exertion by H2O2 generation, and another is indirect affection via NO production. It has been shown that H2O2 induces eNOS via a Ca2+/calmodulin-dependent protein kinase II/janus kinase 2-dependent pathway (Cai et al., 2001). We also found in this study that H2O2 not only up-regulates eNOS, but also produces NO, suggesting a plausible dependence of H2O2-induced antioxidation on NO signaling.

Nevertheless, how NO attenuates oxidative stress remains inclusive. It has been previously described that NO can non-covalently bind to COX, leading to the reversible COX inhibition and transient respiratory dysfunction (Mason et al., 2006). We assumed that an interaction of NO with COX should block electron transport, elicit ROS burst, and trigger antioxidative responses. Our quantification results of antioxidant enzyme quantities as well as the measurement data of ROS levels have altogether confirmed that CR and mimetics allow enhanced antioxidative ability and attenuated oxidative stress in mice. Another consequence of electron transport interruption is a short-term decrease of ATP due to mitochondrial uncoupling and a feedback increase of ATP upon mitochondrial biogenesis. Indeed, we detected the remarkable elevation of ATP levels in the skeletal muscles of mice treated by CR mimetics.

As to the debating issue of NO-mediated mitochondrial biogenesis, we also provided new supporting testimony by showing considerable mitochondrial propagation in the skeletal muscle cells of mice treated by CR mimetics. The discrepancy of findings that CR-mediated lifespan extension with or without mitochondrial biogenesis may be resulted from the earlier or later stages, in other words, an acute short-term or a chronic long-term CR. We have suggested a mechanistic model of dual-phase responses to CR exposure in yeast, in which the phase of mitochondrial enhancement within hours is a respiratory burst phase, and the phase of post-mitochondrial enhancement within days and months is a respiratory decay phase (Wang et al., 2015). It is reasonable that respiratory burst may be attributed to mitochondrial biogenesis, whereas respiratory decay should not be accompanied with mitochondrial biogenesis.

CR is evident to trigger NO production upon Akt-mediated eNOS activation in mice (Nisoli et al., 2003; Nisoli et al., 2005; Cerqueira, Laurindo & Kowaltowski, 2011). ARG is also shown to enhance eNOS expression (Ou et al., 2010), but how ART and SNP affect eNOS remains uncertain. We found in this study that ART, SNP, and ARG not only activate eNOS, but also produce NO, suggesting that they act as CR to initiate NO signaling in mice. A previous investigation has revealed that increase of the AMP/ATP ratio activates AMPK (Anderson & Weindruch, 2010). Earlier evidence has been filed that AMPK activates eNOS through the signaling cascade AMPK→ Rac1→ Akt→ eNOS (Levine, Li & Michel, 2007), in which AMPK and Akt are coordinately responsible for the activation of eNOS through the phosphorylation of Ser1177 (Chen et al., 1999; Dimmeler et al., 1999). Conclusively, ART, SNP, ARG, and H2O2should activate eNOS along AMPK→ Rac1→ Akt→ eNOS.

Because SNP is an NO donor, and ARG is an NO precursor, it should be reasonable that SNP-released NO and ARG-produced NO in vivo can initiate NO signaling. However, ART neither releases NO as SNP, nor produces NO like ARG, so why would it also up-regulate eNOS? It has been indicated that ART alkylates the prosthetic heme of hemoproteins (Zhang & Gerhard, 2009). Our previous work has also shown that ART promotes NO generation by inhibiting the hemoprotein NOS via conjugating the heme moiety and inducing NOS expression either in human cell lines (Zeng & Zhang, 2011) or in bacteria (Zeng et al., 2011). Therefore, we anticipated that ART may interfere with the activity of the mitochondrial hemoprotein COX in direct and indirect ways: it may conjugate to the heme moiety of COX to repress its function; and it may also firstly conjugate to the heme moiety of eNOS to inhibit its activity, and secondly induce the overexpression of eNOS for NO generation and COX binding.

To verify those two possibilities, we should confirm the synchronous up-regulation of COX and eNOS upon induction by ART, thereby validating ART-COX and ART-eNOS interactions. As shown in our results, ART can simultaneously up-regulates eNOS and COX4, suggesting that the inhibition of eNOS and COX by ART may lead to the induction of eNOS/NOS3 and COXexpressions. Although the adducts of ART with hemoproteins were identified in human cell lines, bacteria, and yeast (Zeng & Zhang, 2011; Zeng et al., 2011; Wang & Zeng, 2014), we are at present unable to discriminate the ART-eNOS adducts from the ART-COX adducts or other ART-hemoprotein adducts.

It is worthy of indicating that low-dose ART is found, for the first time, to simulate the lifespan-prolonging effect. In regard to the dose-effect issue of ART, an earlier pharmacokinetical research indicated that when ART was administered at a dose of 6.7 mg/kg, a peak level of 0.82 µg/ml was attained in mice after four hours. This is a concentration more than 5000 times the IC50 of ART in the in vitro tests on Plasmodium berghei for antimalarial activity, and is also close to the human exposure that we see with clinical doses of ART (Zhao et al., 1989). Therefore, we used quite a low dose (50 µl 260 µM or 0.25 mg/kg) of ART for telomere protection in mice. We choose the dose of ART in the present study because we have previously used a similar dose of ART (100 µl 60 µg/ml or 0.3 mg/kg) for NOS induction and NO production in mice (Bao et al., 2012; Wu et al., 2012).

In this study, we noticed that CR or mimetics leads to the global down-regulation of many ubiquitylation pathway genes including DNA repair genes, such as BRCA1, BARD1, and TRP53, implying a cause-result relationship between rare DNA damage and less DNA repair. Furthermore, we also observed that some BRCA1 partners are down-regulated or unchanged in CR and mimetics-treated mice, strengthening a reverse relevance of DNA repair to DNA damage. BRCA1 is structurally identified to interact with other partner proteins for DNA repair (Clark et al., 2012), during which BRCA1 is recruited to the telomere and regulate telomere length and stability, in part through its presence at the telomere (Ballal et al., 2009). BRCA1 and BARD1 constitute a heterodimeric RING finger complex with ubiquitin ligase (E3) (Hashizume et al., 2001). A conclusion of repressed protein degradation is supported by the findings that CR significantly reduces age-related impairments in proteasome-mediated protein degradation, and inhibits age-related increases in ubiquitinated, oxidized, and sumoylated proteins (Li et al., 2008).

Telomeres have been recently verified to be a favored target of persistent DNA damage in aging and stress-induced senescence (Hewitt et al., 2012), and a reverse correlation of BRCA1 with TERT has been previously established (Xiong et al., 2003), implying that the down-regulation of DNA repair genes is an important hint indicating attenuated DNA damage due to ROS scavenging by inducible antioxidation. Indeed, we detected longer telomeres in mouse skeletal muscle cells among treatment groups than those in AL mice. However, whether longer telomeres are due to compromised telomere shortening or enhanced telomere extension is unclear. Our preliminary results on the amplification of TERT mRNA show that it is unlikely up-regulated after underlying treatments. At the same time, we also observed the co-localization and overlap of BRCA1 and TERT with almost identical abundance, implying that TERT is synchronously fluctuated with BRCA1. Actually, we have testified the down-regulation of BRCA1 in RT-PCR array, so it is likely that longer telomeres are attributed to less DNA damage due to mitigated telomere shortening rather than more DNA repair leading to active telomere extension.

In conclusion, we revealed the mechanistic episodes of the effects of CR and mimetics on the dynamic changes of telomeres in mouse skeletal muscle cells. We also provide the direct information supporting the hormesis hypothesis by the validation of beneficial roles of CR mimetics on DNA protection. Therefore, our study should shed light on the discovery of new targets and development of new anti-aging drugs towards longevity.

Supplemental Information

Table S1 Microarray of 84 ubiquitylation pathway genes in mouse skeletal muscles treated by ART

Fold changes were calculated from the comparison of ART with AL.

Click here for additional data file.

Table S2 Microarray of 84 ubiquitylation pathway genes in mouse skeletal muscles treated by SNP

Fold changes were calculated from the comparison of SNP with AL.

Click here for additional data file.

Table S3 Microarray of 84 ubiquitylation pathway genes in mouse skeletal muscles treated by ARG

Fold changes were calculated from the comparison of ARG with AL.

Click here for additional data file.

Table S4 Microarray of 84 ubiquitylation pathway genes in mouse skeletal muscles treated by H2O2

Fold changes were calculated from the comparison of H2O2 with AL.

Click here for additional data file.

Table S5 Microarray of 84 ubiquitylation pathway genes in mouse skeletal muscles treated by CR

Fold changes were calculated from the comparison of CR with AL.

Click here for additional data file.

We thank Kangchen Biotechnology Co, Shanghai, China for performance of RT-PCR array experiments. We also thank Ms. Zeng Weixia in Laura Biotechnology Co, Guangzhou, China for her technical assistance in TRF analysis.

Additional Information and Declarations

Competing Interests

Author Contributions

Animal Ethics

Microarray Data Deposition

The authors declare there are no competing interests.

Da-Ting Wang performed the experiments, analyzed the data, contributed reagents/materials/analysis tools, prepared figures and/or tables, reviewed drafts of the paper.

Jiang He performed the experiments.

Ming Wu, Si-Ming Li and Qian Gao performed the experiments, reviewed drafts of the paper.

Qing-Ping Zeng conceived and designed the experiments, analyzed the data, wrote the paper, prepared figures and/or tables, reviewed drafts of the paper.

The following information was supplied relating to ethical approvals (i.e., approving body and any reference numbers):

All animal procedures were in accordance with the animal care committee at the Guangzhou University of Chinese Medicine, Guangzhou, China. The protocol was approved by the Animal Care Welfare Committee of Guangzhou University of Chinese Medicine (Permit Number: SPF-2011007).

The following information was supplied regarding the deposition of microarray data:

GEO: http://www.ncbi.nlm.nih.gov/geo/query/acc.cgi?acc=GSE65993.

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
