# Peer review of "Artemisinin mimics calorie restriction to trigger mitochondrial biogenesis and compromise telomere shortening in mice"

_PeerJ, doi:10.7717/peerj.822_

## Round 0.1 · original submission · Major Revisions

In particular, you have to clarify the methodology and the English needs revision.

·

Basic reporting

This is OK.

Experimental design

Well designed.

Validity of the findings

The authors have provided sufficient data to support the conclusions. The statistical analysis was OK. The findings appear valid and convincing.

Additional comments

Some studies have evidenced the toxicity of artesunate at different doses, even at the therapeutic dose.There is need to explain how the present dose relates to the therapeutic dose of artesunate.

It is also important to relate the current findings to the reported toxicity of artesunate.

Since artesunate is also taken as combination therapy with amodiaquine, it would be interesting to know if the same conclusions can be obtained with artesunate-amodiaquine combination therapy. The author may wish to investigate this later, or recommend this for future studies.

Reviewer 2 ·

Basic reporting

In this paper the authors report the impact of activation of NO production pathways on antioxidant response factors, expression of genes involved in the ubiquitin proteasomal pathway, telomere length, and protein levels of factors involved in regulation of mitochondrial function. The premise is interesting and the antioxidant response and gene expression data look strong. In several places sufficient detail in experimental design and data analysis are lacking, and there is a tendency to overstate the equivalence of short-term treatments and CR. Interventions to enhance NO signaling are expected to match those aspects of CR that activate NO signaling (including established downstream factors) but this is not the same as drawing broader equivalence. The induction of eNOS and the antioxidant response data are strong (Fig 1 and 2), peroxide serves as suitable control. The gene expression data is nicely presented although the authors omitted directing the reader to the supporting tables. The impact of CR and NO signaling activators on the ubiquitin/proteasome pathway is likely important. Gene expression data are evaluated against protein detection by confocal imaging in (parallel?) treated tissues but the images are too dim and key details in the approach are missing. The telomere data are not strong. One specimen per treatment only is shown and it is difficult to understand how long “three times on every two days” is. The use of a time course of response to the treatments throughout is revealing, although there is some concern about appropriate controls for Figure 6. Overall this is a very interesting paper but it will need some additional work including more detail about the approach and I would suggest toning down some of the conclusions drawn.

Experimental design

1. Provide mouse genetic background and ages of mice used in each experiment and for Fig3, Fig4 and Fig5 please specify the duration of treatment. Also indicate which tissues were harvested and how they were processed. Were sequential injections into the same location? Where the data generated from the tissues that were at the site of injection?
2. Please outline the approach used in generation of data in Fig.5. Were fixed tissue sections used? What was the nature of the dissection referred to in the text? The resolution was not high enough for these data to be evaluated fairly. Equivalence of fluorescence signal is not the same as equivalence in levels as suggested in the text – the signal is dependent on antibody sensitivity – comparisons can be made within detection experiments but not among detection experiments. The legend suggests that treatments were “for three times every two days”, it is unclear what duration of the experiment was.
3. Please check for inconsistencies between data and text, on page 4 there is a confusing section on the impact of treatments on SOD, MnSOD, GSH and CAT for example the sentence beginning “except for SOD…”. Also it appears peroxide not ARG is the most effective inducer of CAT. Was COX evaluated by western (methods) or by ELISA (text)?
4. It is unclear in the text that the DNA repair gene data came from the ubiquitylation RT panel and not a microarray as suggested.
5. On page 8 the authors suggest that treatments lead to mitochondrial uncoupling but there is no data to support this. Furthermore, the argument for biogenesis is not supported as mitochondrial density was not assessed. In the discussion the mitochondrial uncoupling comes up again but now as a downstream effect of respiratory chain activation by inhibition of complex IV. Mitochondrial activities were not measured. A clearer demarcation between data presented and speculation based on data published elsewhere is warranted.
6. For table 1 time course data there is considerable variability in the AL values – it may be more helpful to show these data as fold change from AL. It seems odd that the PGC-1a values at 6h were not significant just looking at the raw numbers. Presumably each time point data were normalized against GAPDH but Fig6 shows only one for the entire set. For the figure as a whole were any of these reprobes of blots or were separate blots used for each antibody. How exactly were the data normalized.
7. In the discussion a model is presented of ART inhibition of eNOS as a means to enhance NO production. NO levels were not measured in any of the experiments. The model requires that production of NO (SNO and ARG) and depletion of NO (inhibition of NOS through ART) achieve the same effects. Perhaps this could be clarified, how is the newly produced NOS insensitive to the inhibitory compound that precipitated increased expression?

Validity of the findings

Without some of the key experimental methods and details a thorough and fair evaluation of the validity study is going to be tricky

Additional comments

This is an attractive study that is likely to be of interest to PeerJ readers. My opinion is that it needs a little work to improve readability and to clarify aspects of the experimental design.

---

## Round 0.2 · accepted · Accept

Note that the quality of some figures are not good in the pdf, e.g., fig 3. Please, check the quality of all figures, before the final publication. Thank you again for your contribution to PeerJ.